# The Contribution of Low-Carbon Energy Technologies to Climate Resilience

**Liliana Proskuryakova** 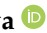

Research Laboratory for Science and Technology Studies, National Research University Higher School of Economics, 101000 Moscow, Russia; lproskuryakova@hse.ru

**Abstract:** The UN vision of climate resilience contains three independent outcomes: resilient people and livelihoods, resilient business and economies, and resilient environmental systems. This article analyzes the positive contributions of low-carbon energy technologies to climate resilience by reviewing and critically assessing the existing pool of studies published by researchers and international organizations that offer comparable data (quantitative indicators). Compilation, critical analysis, and literature review methods are used to develop a methodological framework that is in line with the UN vision of climate resilience and makes it possible to compare the input of low-carbon energy technologies climate resilience by unit of output or during their lifecycle. The framework is supported by the three relevant concepts—energy trilemma, sharing economy/material footprint, and Planetary Pressures-Adjusted Human Development Index. The study identifies indicators that fit the suggested framework and for which the data are available: total material requirement (TMR), present and future levelized cost of electricity (LCOE) without subsidies, $CO_2$ emissions by fuel or industry, lifecycle $CO_2$-equivalent emissions, and mortality rates from accidents and air pollution. They are discussed in the paper with a focus on multi-country and global studies that allow comparisons across different geographies. The findings may be used by decision-makers when prioritizing the support of low-carbon technologies and planning the designs of energy systems.

**Keywords:** greenhouse gas emissions; renewable energy; energy affordability; air pollution; material requirements

## 1. Introduction

There is a growing volume of research publications on energy systems and climate resilience that mainly focus on direct impact of extreme whether phenomena on energy ecosystems [1]. The last decades have witnessed energy transition from fossil fuels to renewable and clean energy sources, from wasteful consumption of natural resources to circular economy and resource efficiency. The fourth major transformation in the energy industry was spurred by the sustainable development and green growth goals, as well as the implementation of international climate agreements [2]. The world's largest economies (The European Union, Japan, Canada, the USA and other) prioritize research and development (R&D) in renewables and other low-carbon solutions. These changes have already led to the creation of a multibillion technology market. Multinational and national energy companies (including oil and gas ones, such as Equinor, Shell, Petrobras, etc.) undertake R&D to develop new low-carbon energy technologies and target this market segment. Moreover, many extractive industry companies are diversifying their business further with a view to become integrated energy companies [3].

While in early 2000s, the structural changes in the global energy industry were less visible, while in 2020 renewables provided around 90% of global capacity growth [4]. According to forecasts made by international organizations and companies (International Energy Agency, International Gas Union, ExxonMobil and Shell companies, and Association of Oil-Exporting Countries), the share of renewables in global energy consumption in 2030

will constitute 14–23%. By 2040, the global demand for all types of energy resources will increase, but the pace of this growth for renewables will be considerably higher (up to 8.9%) than for oil (from 0.5%) [5,6].

The fast growth of renewables occurs due to raising technological efficiency of solar and wind power plants coupled with the capital costs decrease. These two factors already allowed reaching grid parity with traditional heat power plants. Energy storage systems may compensate for variable nature of renewables, and their costs have also been falling. Among the barriers for further advancement of renewables are path dependency of energy companies and energy infrastructure [7], the need to ensure return on investments for heat power plants already in operation, preferences and allowances for exploration and extraction of hydrocarbons [8,9], as well as limited access of the developing countries to modern clean energy technologies [10]. The use of hydrocarbons, particularly natural gas, may also be much more climate neutral [11]. The 2050 final energy consumption will most likely see reductions in oil and coal due to increased energy efficiency, higher share of electricity in final energy consumption, and a significantly higher proportion of renewables [12]. Nuclear power will remain at its current level and range from 6 to 12% compared with 10.5% in 2020 [13].

In this study, the International Energy Agency's definition of low-carbon technologies is used. These are technologies that emit less greenhouse gases (primarily, $CO_2$ and methane) through the entire lifecycle compared with other energy technologies. This study focuses only on low-carbon technologies in the energy industry. Examples of such technologies include power and heat generation from renewables (solar, wind, geothermal), power and heat generation from hydrocarbon fuels together with carbon capture, use, and storage solutions (CCUS), and technologies of hydrogen production and use [14]. This definition is also used by other international institutions such as the International Monetary Fund [15].

The research discourse on low-carbon technologies has been centered on climate mitigation and adaptation. The focus of such studies often rests on a particular country (countries) [16–18], industries [17,19], or technologies [20]. Climate resilience has also been gaining increased visibility with particular focus on energy technologies and energy infrastructure that are expected to form new energy ecosystems [1]. Decarbonizing happens more rapidly in electricity generation than in other sectors, and it is a key element of cost-effective mitigation strategies in most scenarios of the Intergovernmental Panel on Climate Change (IPCC) [21]. Energy technologies have been analyzed in the context of energy systems' resilience to a wide range of threats, of which climate change is only one. These studies point to the significant negative impact of climate on fossil fuel thermal power plants [22,23], small climate impacts on renewable primary energy use that vary by country and type of energy source [24], and inadequate consideration of climate change factors in planning and adapting urban energy systems [25]. In an attempt to capture the progress toward green growth and energy transition, researchers are offering quantitative measurement frameworks that allow comparing countries, energy sources, and the role of various factors, such as social imbalance and governance quality [26]. The studies that offer various indicators to assess energy technologies in the context of climate resilience are rare. Some of these studies do not relate resilience to climate change [27], focus only on specific technologies, such as microgrids [28], or do not focus on the resilience aspects of climate change [29]. Some of these approaches take into account human development aspects or assess the contribution of climate and/or energy technologies to human development [30]. Those studies that assess the contribution of energy technologies to climate resilience, focus on certain climate resilience aspects only, such as material consumption [31] or greenhouse gas emissions [32], and some of them are included in this review.

There are few comprehensive studies that take a multi-aspect approach. The most notable studies were undertaken or commissioned by international organizations—the UN Economic Commission for Europe (UNECE), United States Environmental Protection Agency (US EPA), and Economist Intelligence Unit (EIU). The US EPA study is aimed

at assessing climate resilience at the county level and offers a conceptual approach and a set of indicators for constructing the Climate Resilience Screening Index (CRSI). The CRSI consists of the five thematic blocks—natural environment, built environment, society, governance, and risk [33]. The EIU Climate Change Resilience Index assesses the largest world economies' ability and willingness to confront climate change, and includes eight indicators, of which only two are focused on mitigation and adaptation: adaptation costs and mitigation costs [34]. The UNECE study is conceptually most close to the present review paper and includes a description, lifecycle inventory, and environmental impact assessment of various energy technologies (coal, natural gas, wind, solar, hydro, and nuclear). These are followed by various climate change aspects of these technologies, including resource and material use, land occupation, and others. Many indicators offer technology comparison per 1 kWh of output, but some indicators are expressed in points (for example, lifecycle land use), which limits the objectivity of analysis [35].

This paper aims to review and critically assess the existing pool of studies published by researchers and international organizations that offer comparable data (quantitative indicators) on the contribution of various energy technologies to climate resilience. This thematic focus has not been suggested or explored before. This study adds to the existing literature by offering a methodological approach to the selection of energy technology indicators, for which the data exist, as well as by pointing to research and data gaps. The suggested indicators may serve as an evidence base for decision-makers that plan investments or policy support measures for low-carbon technologies. This review does not attempt to aggregate the suggested indicators into a composite index due to substantial data gaps and numerous options that may be tailored for different stakeholders and fit different goals.

The rest of the paper is structured as follows. Section 2 outlines materials and methods used in this study, including quantitative metrics that reflect the various measures of climate resilience that may be attributable to low-carbon energy technologies. The results outlined in Sections 3.1, 3.2, and 3.3 are focused on the main elements of these metrics—material requirements, present and future energy affordability, and greenhouse gas emissions and air pollution. The results are followed by conclusions (Section 4).

## 2. Materials and Methods

The UN vision of climate resilience contains three independent outcomes: (1) resilient people and livelihoods that include addressing energy affordability to the poorest countries and communities; (2) resilient business and economies that include addressing increased resource (material) use by all industries and related shortages/ competition among them, and (3) resilient environmental systems that include addressing water and air pollution. This approach places the focus on people and nature across different geographies and sectors [36]. This study is aligned with the UN Resilience Action category 6 "Climate-proofing of infrastructure and services" [36] and adds to the indicators developed by the Glasgow–Sharm el-Sheikh work program on the global goal on adaptation with regard to SDG7-related indicators that are currently missing [37].

The energy trilemma concept and the related World Energy Trilemma Index calculated by the World Energy Council suggest that a clean and just energy transition captures the achievement of three priorities at the same time: security, equity, and sustainability. However, all of them cannot be attained simultaneously to a full extent (Figure 1) [38]. Liu et al. (2022) examined the data for the top ten $CO_2$-emitting countries (1990–2016) and found that the energy trilemma and energy transition simultaneously enhance economic growth and environmental quality in the long run. Moreover, the energy trilemma alone is negatively correlated with environmental quality [39]. This testifies to complex interrelations among the energy trilemma components and the environment.

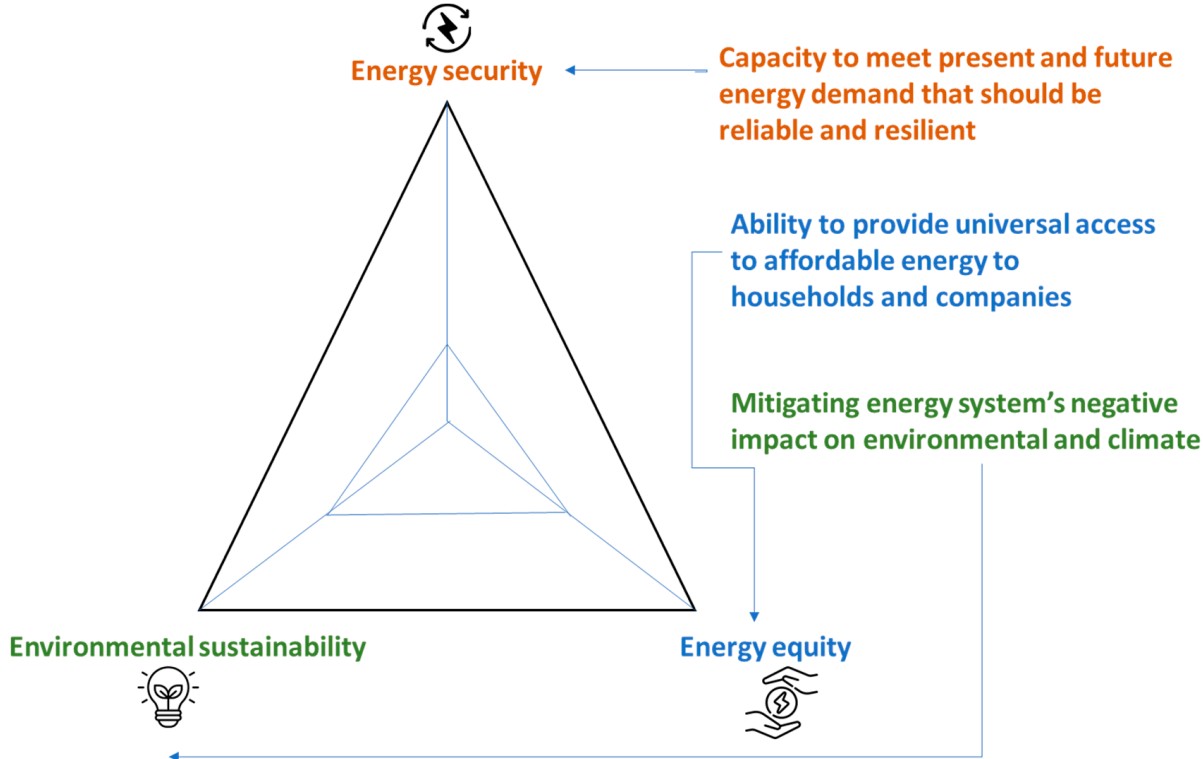

**Figure 1.** The World Energy Trilemma.

Material consumption in all segments of the energy industry has been growing. The most obvious solution has been the advancement of a circular economy that implies green design and manufacturing, as well as reuse, repair, and post-consumption (Figure 2). Multiple benefits are brought by the sharing economy that assures synergy between technology innovation, information, knowledge, entrepreneurship, and marketing and promotes more efficient resource use by customers though access (not necessarily ownership) to energy technologies [40]. The sharing economy in the energy industry has also been growing and implies new business models, such as energy-as-a-service [41]. The material footprint of renewables is different from fossil fuels: while renewables are commonly seen as less material-intensive, some studies argue that renewable energy does not contribute to dematerialization [42]. The most promising material demand scenarios rely on future less material-intensive technologies and large-scale recycling [43].

The Human Development Index (HDI) developed by the United Nations Development Program measures progress in three main dimensions of human development: life expectancy at birth, years of schooling, and Gross National Income (GNI) per capita. The HDI is interlinked with clean energy and researchers have explored those connections. For instance, a study has identified a unidirectional causality relationship from the HDI to renewable energy consumption, and to research and development expenditure [44].

The Planetary-Adjusted Human Development Index (PHDI) discounts the HDI for pressures on the planet. It is computed as the product of the HDI and the index of planetary pressures, where latter can be seen as an adjustment factor. In other words, the PHDI is the level of human development adjusted by $CO_2$ emissions per person (production-based) and the material footprint per capita to account for the excessive anthropogenic pressure on the planet. In an ideal situation, there are no pressures on the planet, and the PHDI = the HDI. In the real world, as pressures increase, the PHDI < the HDI (Figure 3) [45].

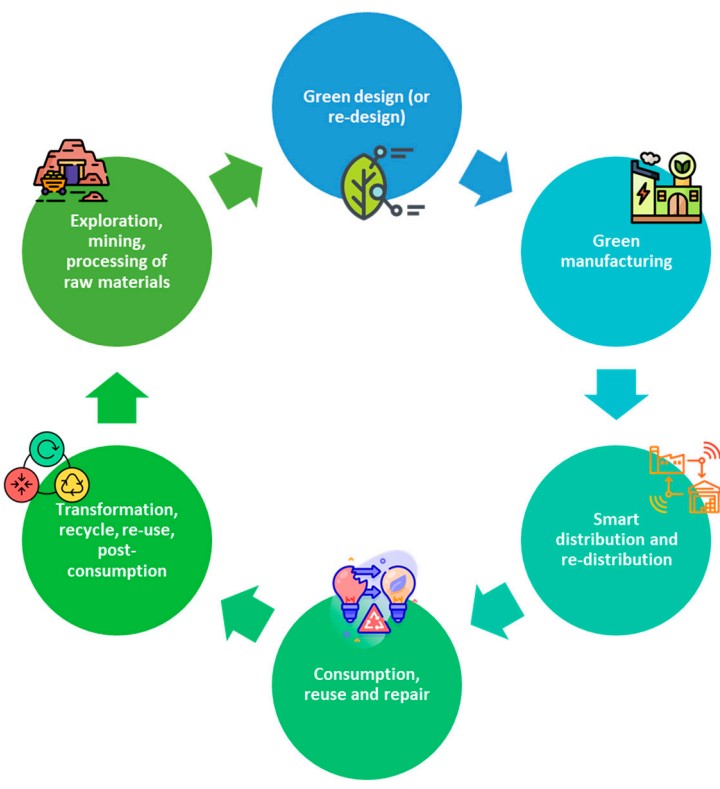

**Figure 2.** Circular economy for the energy industry.

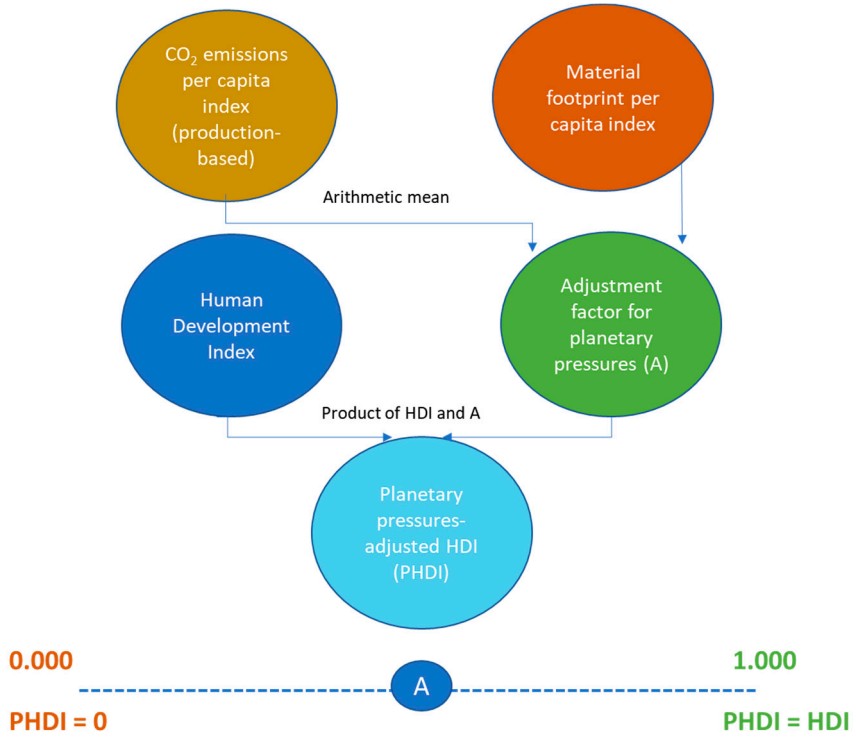

**Figure 3.** Construction of Planetary-Adjusted Human Development Index.

If applied to the energy industry, the UN approach to climate resilience may be supported by the concepts described above—energy trilemma, sharing economy/material footprint, and PHDI. The present review paper focuses on the following components of climate resilience: present and future energy affordability, GHG emissions and environ-

mental pollution, and material footprint. Additionally, the human development focus and global data coverage are assured (Figure 4).

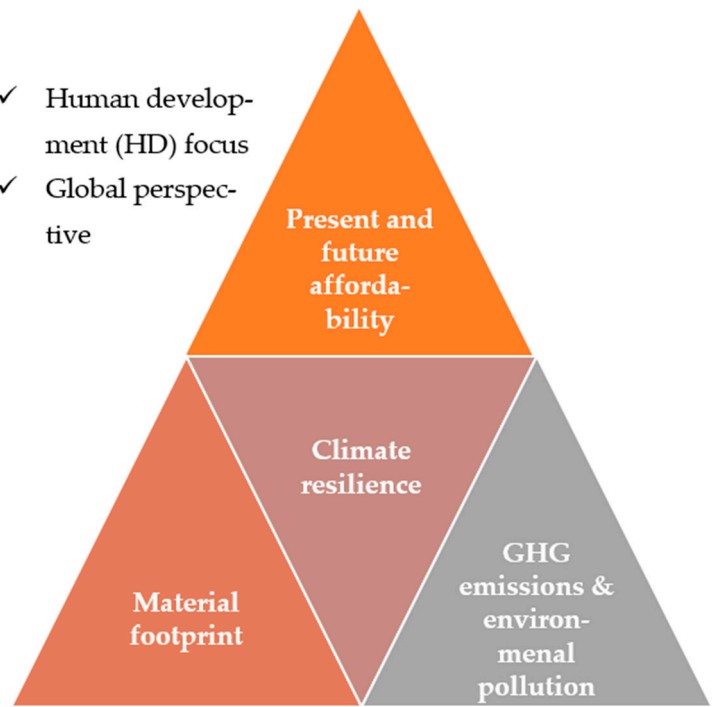

**Figure 4.** The approach to climate resilience applied in this study.

The described methodological framework is further developed with the use of compilation, critical analysis, and literature review methods. Literature review was conducted in order to synthesize the scholarly publications within the thematic scope summarized by Figure 4. This method is the main one applied in review studies [46]. Compilation and critical analysis of selected publications were undertaken to identity data and information gaps and draw conclusions that may be useful for researchers and practitioners looking for state-of-the art evidence to guide their decision-making and work practices [47].

Based on the described approach, an initial search for data and quantitative studies was undertaken in ScienceDirect, Google Scholar, and international organizations' databases. Different keywords were used for information and data related to different indicators; they are summarized in Supplementary Materials (Table S1). The search for keywords in research publications was undertaken in titles, abstracts, or author specified keywords. Whenever the search for research papers yielded more than 200 entries, the search was narrowed down to review papers. In most cases, scanning through articles' and book chapters' names was sufficient to identify those that cover multiple energy sources and take a global perspective and, thus, fit the research approach of this study. The selection was based on the principles outlined in Figure 4 and the availability of data that makes it possible to compare the different low-carbon technologies across geographies and sectors. The technologies may be compared by those indicators that reflect their emissions, availability, and material requirement per unit of output or during the entire lifecycle.

## 3. Results

The articles and datasets that were identified through the initial information search made it possible to select the following indicators that reflect the three aspects of the UN climate resilience vision: (1) the material footprint; (2) the present and forecasted levelized cost of electricity (LCOE) without subsidies; (3) mortality rates from accidents and air pollution per unit of electricity worldwide by energy source; $CO_2$ emissions by fuel or industry; lifecycle $CO_2$-equivalent emissions and mortality rates from accidents and air pollution

per unit of electricity worldwide by energy source (Table 1). The studies that analyze these aspects and calculate the selected indicators are reviewed in the subsequent subsections.

**Table 1.** Indicators that reflect the three aspects of the UN climate resilience vision.

| UN Vision of Climate Resilience Components | Aspect Selected for Analysis | Indicators | Based on |
|---|---|---|---|
| Resilient business and economies that include addressing increased resource (material) use by all industries and related shortages/competition among them | Material requirement | • Total material requirement (TMR) | • PHDI<br>• Circular/sharing economy |
| Resilient people and livelihoods that include addressing energy affordability to the poorest countries and communities | Present and Future Energy Affordability | • Present and future levelized cost of electricity (LCOE) without subsidies | • The World Energy Trilemma |
| Resilient environmental systems that include addressing water and air pollution | GHG emissions Health implications of air pollution and accidents | • $CO_2$ emissions by fuel or industry<br>• Lifecycle $CO_2$-equivalent emissions<br>• Mortality rates from accidents and air pollution per unit of electricity worldwide by energy source | • PHDI<br>• The World Energy Trilemma |

Source: author's analysis.

This section is structured in line with the three aspects of climate resilience selected in this study: (1) the material footprint, (2) the present and future energy affordability and (3) the GHG emissions and health implications of air pollution. The focus is on datasets and quantitative studies that offer comparable data and a multi-country perspective.

*3.1. Material Requirement*

The total material requirement (TMR) coefficient is a measure of all physical materials associated with the entire lifecycle of various types of power, including "hidden" materials such as processing waste and soil erosion. For instance, for nuclear power, this includes mining and milling of uranium, its enrichment and fuel production, reactor construction and operation, fuel reprocessing, reactor decommissioning, and spent nuclear fuel disposal. Another example: solar PV panel manufacturing requires silver (processed into silver paste and c-Si cell manufacturing). Copper, indium, gallium, selenium (processed into CIGS powder), silica (processed into glass), cadmium, and tellurium (processed into CdS powder and CdTe powder) are used for CIGS and CdTe panel manufacturing. Aluminum is used for frames. All of these materials and products that are subsequently used for assembling a PV module, while copper is used to manufacture BOS at the stage of PV system assembly [43].

There are relatively few studies that assess the TMR by energy technology; a number of studies aggregate data to assess the TMR for the entre energy transition by sector (power generation), by type of materials, or by scenario. With regard to nuclear, it was found that resource use differs substantially depending on the type of nuclear fuel mining and nuclear power plant: closed-cycle power plants are 26% less resource intensive than open cycle, and in situ leaching (ISL) is the most resource-efficient of all mining methods. The combination of ISL and closed-cycle power generation allows the TMR to lower to almost 0.1 kg/kWh, whereas for other options it can go to above 0.5 kg/kWh. As shown by Nakagawa et al., the TMR coefficient is highest for coal and oil, rather high for LNG, and comparably low for nuclear (For comparing the TMR for nuclear power generation, the following assumption was made by the authors of the study: 25% open pit, 25% underground, and 50% in situ

leaching (ISL), with 50% open cycle and 50% closed cycle) and solar PV generation. Nuclear is 20% lower that of coal power generation, 23% that of oil power generation, and 35% that of LNG power generation [48].

According to a recent study, the existing deposits of aluminum, steel, and rare earth metals are sufficient to power green energy transition (Table 2). However, several-fold expansion of global production will be required for Nd, fiberglass, Dy, solar-grade polysilicon, and Te. The highest demand will be for bulk materials—aluminum, copper, and steel that are required by all types of generation. Additionally, there will be high demand for indium, selenium, gallium, manganese, nickel, and glass. If carbon capture and storage are applied for natural gas, coal, and biomass generation, the requirement for most materials will grow, but not very significantly. Government policies are required to stimulate recycling and technological innovation to reduce material demand, as well as promote responsible mining [31].

**Table 2.** Material requirements by renewable and other energy technologies.

| | Conventional Solar | Thin-Film Solar | Onshore Wind | Offshore Wind | Other Technologies * |
|---|---|---|---|---|---|
| Aluminum | X | X | X | X | X |
| Cadmium | | X | | | |
| Cement | X | X | X | | X |
| Copper | X | X | X | X | X |
| Dysprosium | | | X | X | |
| Fiberglass | | | X | X | |
| Neodymium | | | X | X | |
| Polysilicon | X | | | | |
| Silver | X | | | | |
| Steel | X | X | X | X | X |
| Tellurium | | X | | | |

Source of data: [31], Note: * other technologies include fossil fuels, geothermal, hydroelectric, and nuclear.

According to UNECE assessments, solar power is one that requires more minerals, materials, land, and water compared to wind and hydropower. Particularly high are indicators for poly-Si, roof-mounted, photovoltaic power production technologies. Hydropower has the lowest resource-consumption indicators per 1 kWh of generated electricity compared to solar and wind. Solar PV installations show the highest indicators of minerals and metals intensity as well as dissipated water and freshwater eutrophication, while land use is highest for CSP technologies [35].

The World Nuclear Association offers a snapshot of critical minerals required for different generation technologies per MW of installed capacity (Figure 5) and per unit (TWh) of electricity (Figure 6). According to this data, offshore and onshore wind substantially bypass other types of power generation, especially in copper and zinc. Wind generation is followed by solar PV, nuclear, and fossil fuels [49].

Of the published studies found on the topic, only a small number look into the future and attempt to project changes in material intensity of low-carbon technologies that may occur due to technological improvements. Even though the majority of studies are global in their scope, they only focus on a specific material of technology [50]. There is a role of sharing economy in reducing material footprint, as it makes a meaningful contribution to increasing energy efficiency and sustainable production and consumption in the energy industry [51].

### 3.2. Present and Future Energy Affordability

The levelized cost of energy is a measure used by several international organizations and in different studies. It makes it possible to compare the average net cost of different types of electricity generation (Tables 3 and 4), energy storage, and even hydrogen production (Table 5) over the entire lifetime [52,53]. Therefore, this is a fairer expression of

an energy generation cost as compared with capital costs, net present value, and other economic indicators. It is expressed as total annual costs divided by the total annual output [54].

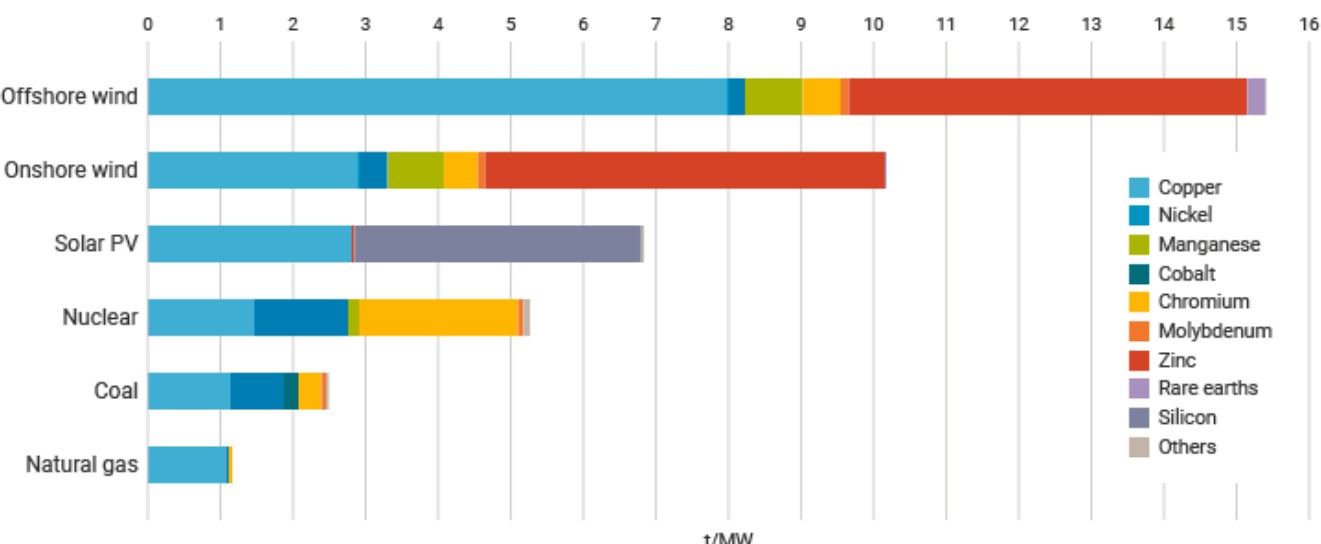

**Figure 5.** Critical minerals requirement for different power generation technologies per MW of installed capacity, Source of data: [49].

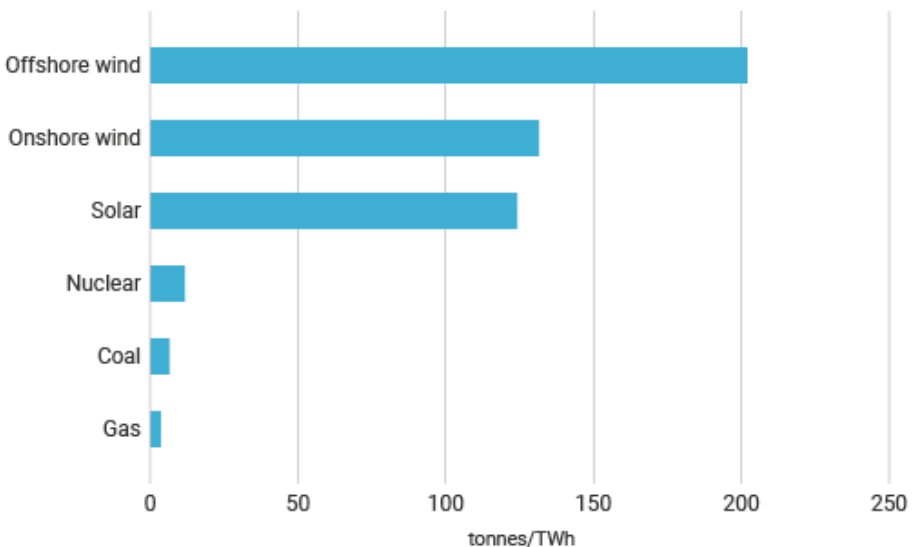

**Figure 6.** Critical minerals requirement for different power generation technologies per TWh of electricity generated, Source of data: [49].

Assessments of off-grid renewable energy systems in 161 countries indicate that inflation-adjusted LCOEs range from 0.03 $\$_{2021}$/kWh in Saudi Arabia to 0.99 $\$_{2021}$/kWh in Pakistan, with a total mean value of around 0.35 $\$_{2021}$/kWh (median is 0.29 $\$_{2021}$/kWh). These values for hybrid (renewables + fossil fuels) and 100% renewable energy systems on average decreased by 4% and 9% per annum, respectively, between 2016 and 2021 [55]. Due to the fast changes in economic and technological characteristics of renewables and the long-term character of sustainability goals, especially those related to assuring affordable clean energy for all, it is important to look at the projected LCOE of renewables (Table 3) [56].

**Table 3.** Levelized cost of energy comparison—unsubsidized estimations (USD/MWh)—Renewables and battery storage.

| Technology | Capacity Factor | Min LCOE | Max LCOE | 2050 Outlook |
|---|---|---|---|---|
| Solar PV-Rooftop Residential | | 117 | 282 | Data unavailable |
| Solar PV-Rooftop Commercial and Industrial | | 67 | 180 | Data unavailable |
| Solar PV-community | | 49 | 185 | Data unavailable |
| Solar PV-Crystalline Utility Scale | | 24 | 96 | The LCOE for utility-scale solar PV will lower from 40 to 10–20 USD/MWh in 2050 in all scenarios. |
| Solar PV-Thin Film Utility Scale Solar standalone (fixed-axis) | 29% | 28–30 | 37–49 | |
| Solar PV + Storage * Solar PV + Storage Utility Scale | 26–28% | 43 46 | 68 102 | Utility-scale PV + battery will follow the same pattern decreasing from above 60 to below 40 in the Conservative scenario and nearly 20 in the Advanced scenario. Conservative scenario envisages expansion of present-day technologies with few innovations. These technologies become increasingly accessible due to continued industrial learning, while public and private R&D decreases. |
| Solar Thermal Tower with Storage | 54.2% | 126 | 156 | The LCOE for concentrated solar power (CSP), unlike most other renewables, is projected to experience stagnation after 2030 in all scenarios. In the Conservative scenario, the LCOE for CSP will not change from 2020 to 2050, while in Moderate and Advanced scenarios there will be a significant drop in 2020–2030 to 40–50 USD/MWh. |
| Geothermal * | 90% | 36–61 | 42–102 | The levelized cost of geothermal energy will also change little after the year 2030 (around 40 USD/MWh in Advanced scenario) in all scenarios, while the price drop in 2020–2030 will be less noticeable than that of CSP. |
| Wind onshore * | 41–43% | 24–30 | 66–75 | The LCOE of land-based wind is projected to decrease from around 30 to almost 10 USD/MWh in 2050 in the Advanced (innovative green) scenario *. By 2040, the LCOE of distributed wind (DW) will lower more significantly to 10–20 USD/MWh for commercial, large, midsize, and residential DW. The dynamics will be most vivid for residential DW, the cost of which in 2020 was near 100 USD/MWh. +Advanced scenario foresees market success of currently new technologies that are not yet on the market. New technologies entail innovative technology architectures that become possible due to increased public and private R&D. |
| Wind offshore * | 44% | 72–110 | 140–170 | The LCOE of offshore wind will follow the same trend and drop twofold to around 40 USD/MWh in 2050 in Moderate and Advanced scenarios. |

**Table 3.** *Cont.*

| Technology | Capacity Factor | Min LCOE | Max LCOE | 2050 Outlook |
|---|---|---|---|---|
| Hydroelectric * | 54% | 48.96 | 82.65 | Hydropower LCOE will remain almost unchanged in 2020–2050 (around 65 USD), showing only an insignificant decrease in the Advanced scenario after 2030. Same trajectories are portrayed for pumped storage hydropower that will decrease insignificantly only in the Advanced scenario. |
| Battery Storage * | 10% | 114.70 | 141 | Utility-scale battery storage will not decrease after 2030 in the Conservative scenario, while Moderate and Advanced scenarios preview more than a threefold decrease in CAPEX to 500–600 USD/MW. |

Notes: * Advanced scenario foresees market success of currently new technologies that are not yet on the market. New technologies entail innovative technology architectures that become possible due to increased public and private R&D. Moderate scenario previews widespread adoption of present-day cutting-edge innovative technologies. The level of innovation activity remains high within the time horizon, and the existing level of public and private R&D is maintained. Estimations for new resources entering service in 2027 (USD2021 per MWh), sources of date: [6,57–60].

**Table 4.** Levelized cost of energy comparison—unsubsidized estimations (USD/MWh)—Non-renewables.

| Technology | Capacity Factor | Min LCOE | Max LCOE | 2050 Outlook |
|---|---|---|---|---|
| **Gas Peaking** | | **115–151** | **196–221** | **Data Unavailable** |
| Nuclear<br>Advanced nuclear * | 90% | 83–141<br>131 | 99–221<br>204 | It is forecasted that the nuclear LCOE will level off at USD 60–110 after the year 2030 in the Net-zero scenario. |
| Coal | | 65–68 | 152–166 | Coal CCUS will not drop after 2040 and will amount to 77–107 by the year 2050 in the Sustainable Development scenario. |
| Ultra-supercritical coal | 85% | 65–152 | 74–101 | Coal CCUS will not drop after 2040 and will amount to 77–107 by the year 2050 in the Sustainable Development scenario. |
| Combined cycle * | 87% | 34–39 | 50–74 | The LCOE for gas CCGT will amount to USD 60–150 in the Net-zero scenario.<br>Gas CCUS will not change significantly after 2030 and by 2050 will amount to USD 53–118 in the Sustainable Development scenario. |

Notes: estimations note for new resources entering service in 2027 (2021 dollars per MWh). The solar hybrid system is a single-axis PV system coupled with a four-hour battery storage system. Costs are expressed in terms of net AC (alternating current) power available to the grid for the installed capacity. * Advanced scenario foresees market success of currently new technologies that are not yet on the market. New technologies entail innovative technology architectures that become possible due to increased public and private R&D. Moderate scenario previews widespread adoption of present-day cutting-edge innovative technologies. The level of innovation activity remains high within the time horizon, and the existing level of public and private R&D is maintained. Sources of date: [6,57–60].

At present, the most expensive renewables are solar PV-rooftop and community-level installations, solar thermal tower with storage, and wind offshore. Comparing the 2050 LCOE projections for renewables, it may be noted that utility-scale solar and wind power will likely be most cost competitive compared to other technologies. While the levelized cost of most renewables is expected to drop, the cost of hydropower will remain almost unchanged in the coming 30 years. Similarly, the cost of battery storage will not lower after 2030 in the Conservative scenario, while other scenarios preview a several-fold drop in capital expenditure that constitutes the bulk of total costs.

**Table 5.** Levelized cost of energy comparison—unsubsidized estimations—Hydrogen.

| | Alkaline | | | PEM | | | 2050 Outlook |
|---|---|---|---|---|---|---|---|
| Capacity, kW | Small 1000 | Medium 20,000 | Large 100,000 | Small 1000 | Medium 20,000 | Large 100,000 | |
| Hydrogen LCOE (USD/kgH$_2$) | | 1.60–1.90 | 1.40–1.75 | 2.75–2.90 | 2.15–2.40 | 1.90–2.15 | <2 methane reforming and grid-connected electrolysis (NAM) |
| Green Hydrogen | 2.10–2.45 | 3.79–5.28 | 0.83–2.83 | | 4.77–7.37 | 1.68–4.28 | <2 grid-connected electrolysis (EUR and SEA) ≥2 Dedicated solar PV electrolysis and dedicated onshore wind electrolysis (NAM, EUR, SEA, MEA) |
| Pink Hydrogen | | 2.75–4.08 | 0.48–1.81 | | 3.47–5.29 | 1.16–2.99 | |
| Natural gas-equivalent cost (USD/MMBTU) | 18.45–20.50 | 14.05–16.70 | 12.30–15.35 | 24.15–25.45 | 18.90–21.05 | 16.25–18.90 | n/a |
| Natural gas price (USD/MMBTU) | | | | | | | 5 in NAM; 13.5 in EUR 7.6 in MEA 11.4 in SEA |
| Natural gas/Hydrogen blend (USD/MMBTU) * | 6.45–7.06 | 5.57–6.10 | 5.22–5.83 | 7.59–7.85 | 6.54–6.97 | 6.01–6.54 | n/a |

Notes: * Based on 80%/20% blend of natural gas and green hydrogen. Cost of natural gas is 3.45 gas/Hydrogen blend (USD), cost of green hydrogen is based on the natural gas equivalent. NAM—North America; EUR—Europe; MEA—Middle East and North Africa; SEA—South East Asia. Comparable data on 2050 projections for natural gas-equivalent cost of hydrogen was not available from the sources. Sources of data: [6,60,61].

The cheapest of all unsubsidized fossil fuels (Table 4) is natural gas-combined cycle, while the most expensive options are gas peaking and nuclear. The price reductions by 2050 will not be as impressive as for renewables, especially if coupled with CCUS to be in line with climate mitigation plans. The projected 2050 costs of most traditional energy sources are markedly higher than those of renewables, making them less affordable with time.

The LCOE for hydrogen is expected to decrease gradually, but the more precise forecasts are not yet there (Table 5). At present, this indicator is lowest for large alkaline projects—1.40–1.75 USD/MWh. However, compared to alkaline, PEM has a smaller environmental footprint, lower startup and system response times, lower minimum load requirements, and greater load flexibility (that better suit the intermittent renewable energy). Electrolyzer stack costs make up around 33–45% of the total CAPEX, and the cost of electricity makes up approximately 30–60% of the green hydrogen LCOE.

The LCOE of all types of energy resources vary significantly by country. With the initial presumptions of USD/ton 30 carbon price and 7% discount rate, the highest LCOE is assessed for small-scale (<0.014 MW) onshore wind and run of river (<0.25 MW) power plants in Italy. The lowest indicators are registered in Sweden for large nuclear power plants (1000 MW, 20 years) and medium-sized onshore wind projects in Denmark (4.5 MW) [62].

### 3.3. Greenhouse Gas Emissions and Air Pollution

There are several indicators that make it possible to compare low-carbon technologies in terms of their greenhouse gas emissions. In this section, $CO_2$ emissions by fuel or industry and lifecycle $CO_2$-equivalent emissions are reviewed. Additionally, the deathprint and health implications of air pollution and incidents for different power generation technologies are outlined.

### 3.3.1. Carbon Dioxide Emissions by Fuel or Industry

Some studies suggest that reductions in $CO_2$ emissions that occurred in the last decade are attributable to increased productivity of renewable energy sources and the changes in renewable electricity generation per GDP. The significant prerequisite is the energy technology level of a country [32]. Country differences in emissions by fuel type depend on income level (low-income countries emit less), as well as the economic development phase that the country is going through (emerging economies that undergo industrialization emit more). Per capita $CO_2$ emissions are highest for China, Kazakhstan, Mongolia (with an upward trend), South Africa, and Australia (with a downward trend) [63].

Global $CO_2$ emissions from most fuels have been increasing, the largest emissions attributable to coal, biomass co-firing, and oil. Figure 7 features global carbon dioxide emissions by fuel or industry. Emissions from all traditional sources have an upward trend, while others (includes renewables) stay low and flat. The emissions from renewables are negligible and mostly attributable to biomass. Some large hydropower projects may also be a potential source of emission, but this has not been accounted for at the global scale [64].

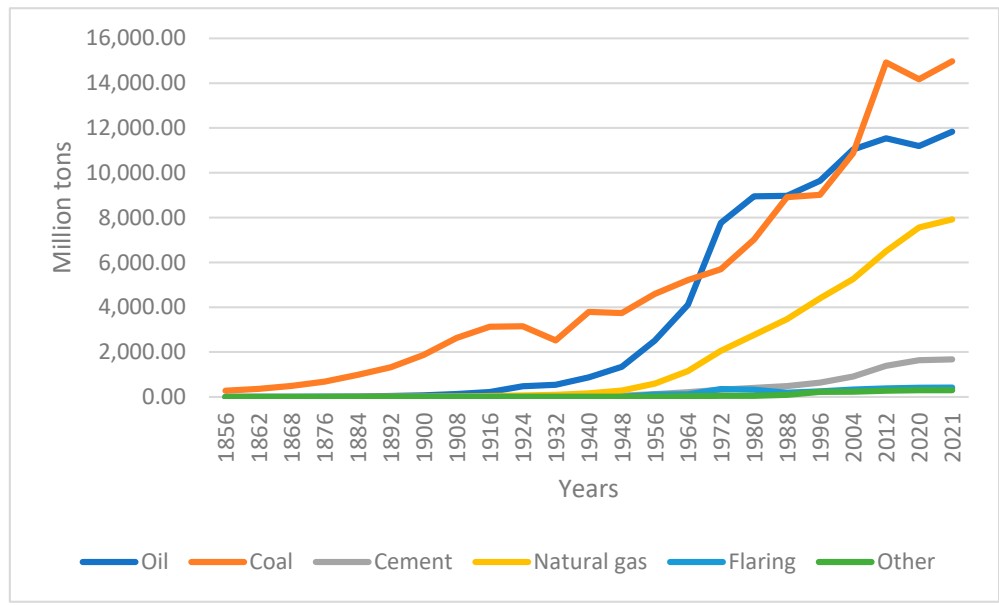

**Figure 7.** Annual global carbon dioxide emissions by fuel or industry from 1856 to 2021, Source: [65].

The 2021–2022 increase in $CO_2$ emissions by fuel is associated with heating and cooling due to extreme weather conditions, disruptions of traditional fuel trade flows, and decline in nuclear power generation, among other. The decrease factors include economic and industrial slowdown and the rapid installation of clean energy power plants [66]. With the exception of China, Brazil, and India, the largest 2022 renewable capacity additions were made by the developed countries—the USA, Germany, Japan, Canada, Spain, France, and Italy. Low-income and least developed countries that are most energy poor were not among the top in this indicator [67]. This testifies to lack of financial resources and/or technologies.

Carbon dioxide emissions from fuel combustion are directly attributable to fossil fuel energy use and thermal power plants. In 2022, this indicator increased again (by 2.5%) to record high above the 2019 level, albeit at a slower pace than in 2021 (Figure 8). In absolute terms, emissions reached 33.8 GtCO$_2$ at the time of global economic downturn. If broken by energy sources, the largest share belongs to coal (46%) followed by oil (32%) and natural gas (23%). In 2022, this indicator was highest for China (10.504 Mt CO$_2$), followed by the USA (4.735), India (2.481), Russia (1.798), and Japan (1.001) [68].

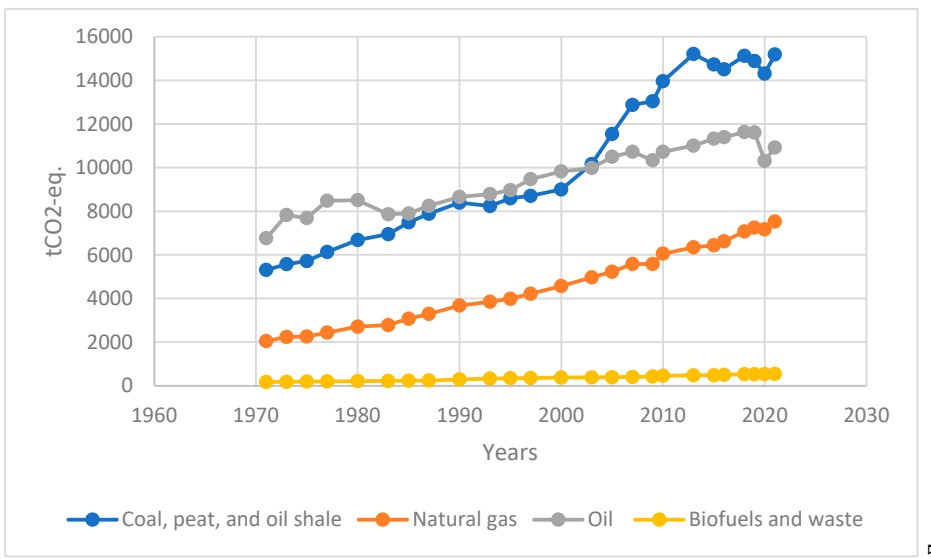

**Figure 8.** Total greenhouse gas emissions from fuel combustion by source from 1971 to 2021. Source: [68].

### 3.3.2. Lifecycle $CO_2$-Equivalent Emissions

Even if compared through the entire lifecycle (construction, operation, and decommissioning), carbon dioxide emissions from renewables are still much lower than those from fossil fuels (Figure 9). The data from UNECE, the Intergovernmental Panel on Climate Change (IPCC), and the World Nuclear Association (WNA) testify in favor of this conclusion. The lowest values belong to wind energy (onshore and offshore) followed by hydropower and solar concentrated.

Comparing lifecycle global carbon dioxide emissions of renewable and non-renewable energy sources with and without CCS, it may be noted that the lowest values are for hydropower, ocean, and wind energy. As the largest emissions are registered at the stage of hydrocarbon use (not mining or transport) [69], CCS could lead to significant reductions in emissions for natural gas, oil, coal, and biomass power plants [41]. Average lifecycle $CO_2$-equivalent emissions per kWh generated at thermal power plants with carbon capture and storage technologies (CCS) are lowest for natural gas compared with other fossil fuels. However, these values are still almost 35% higher than large hydro, the largest $CO_2$ emitting technology among renewables (Figure 9).

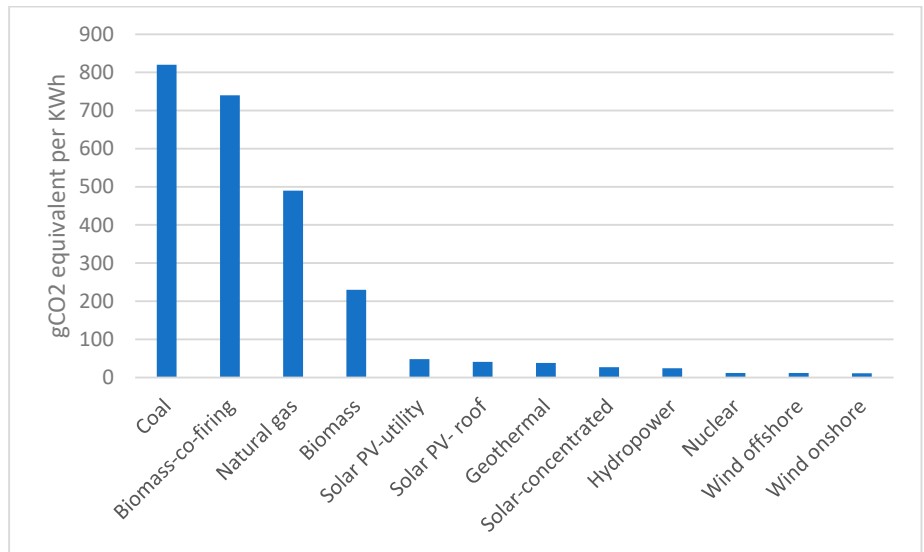

**Figure 9.** Average lifecycle $CO_2$-equivalent emissions. Source of data: [70].

If renewable gases are compared with natural gas, the former possess a significant GHG mitigation potential that may be unveiled if a number of conditions are met, including the use of closed tanks for biomethane, the use of renewable power for power-to-gas solutions, and the extraction of excess heat for bio-SNG [71]. Emissions from nuclear power are at the level of wind (Figure 10). Average emissions from geothermal power plants are comparable to wind energy, though the variations are very substantial: from 3.9 to 1040 g $CO_2$ eq./kWh [72]. Depending on the $H_2$ supply chain selection, hydrogen emissions range from 72 to 746 $gCO_2$/kWh and can be reduced by using renewable electricity instead of grid electricity for hydrogen production [73].

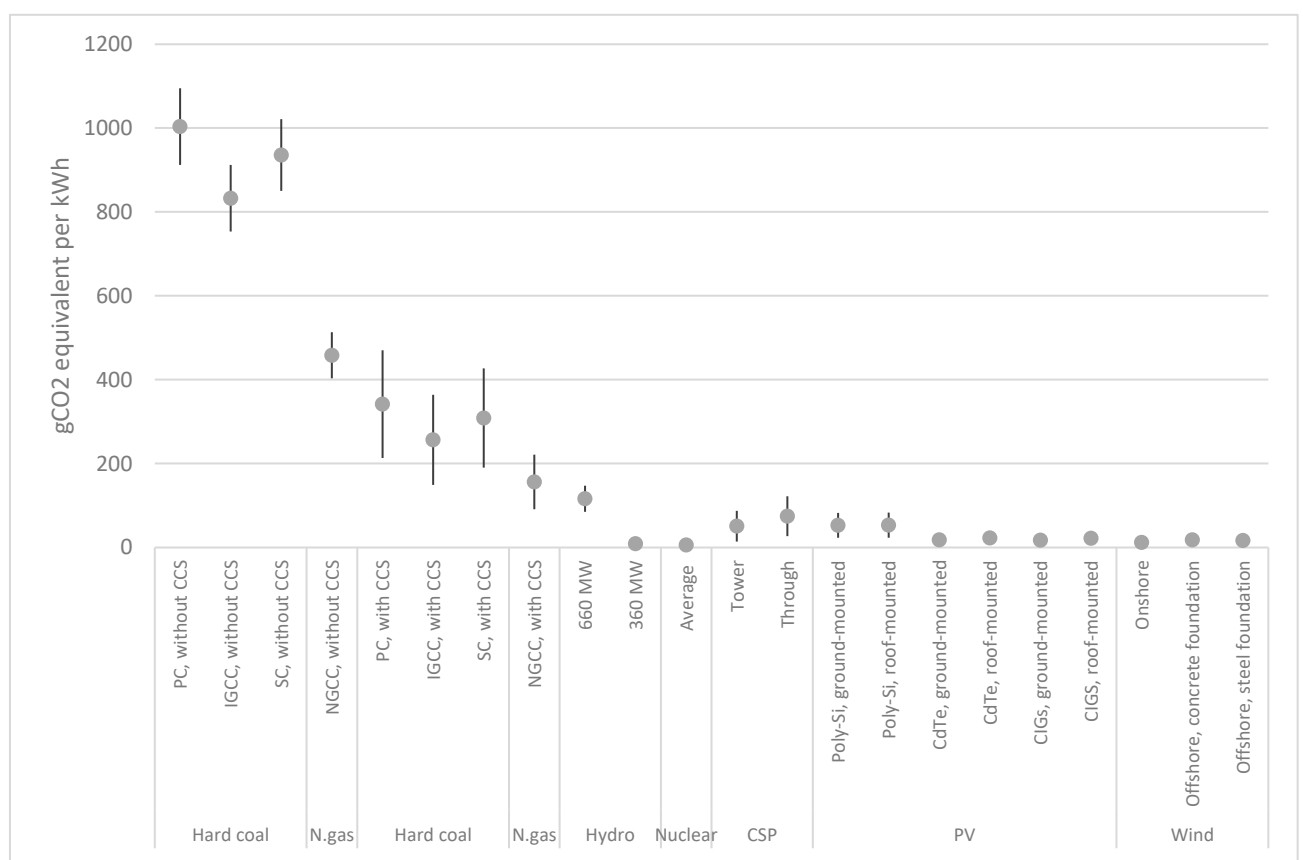

**Figure 10.** Minimum, maximum and average lifecycle $CO_2$-equivalent emissions ($gCO_2$ equivalent per kWh)). Source: [35].

### 3.3.3. Air Pollution and Accidents with Health Implications

Air pollution is harmful for the environment and human health. It is one of the key sources of premature deaths and diseases, such as stroke, heart disease, lung cancer, and various respiratory diseases. A total of 6.7 million premature deaths annually are attributable to ambient air pollution and household air pollution. Death rates from air pollution are highest in low- and middle-income countries due to heavy reliance on solid fuels for cooking and/or industrialization [74]. Over the past decades, the global downward trend in the total air pollution is mainly attributable to rapidly decreasing indoor air pollution [75].

Calculations of mortality rates by energy source makes it possible to compare them by unit of output (Figure 11). It may be observed that hydro, solar, wind, and nuclear are the safest energy sources for the environment and human health. Natural gas is the safest of all fossil fuels. Unfortunately, there are no statistical data for new energy sources like hydrogen.

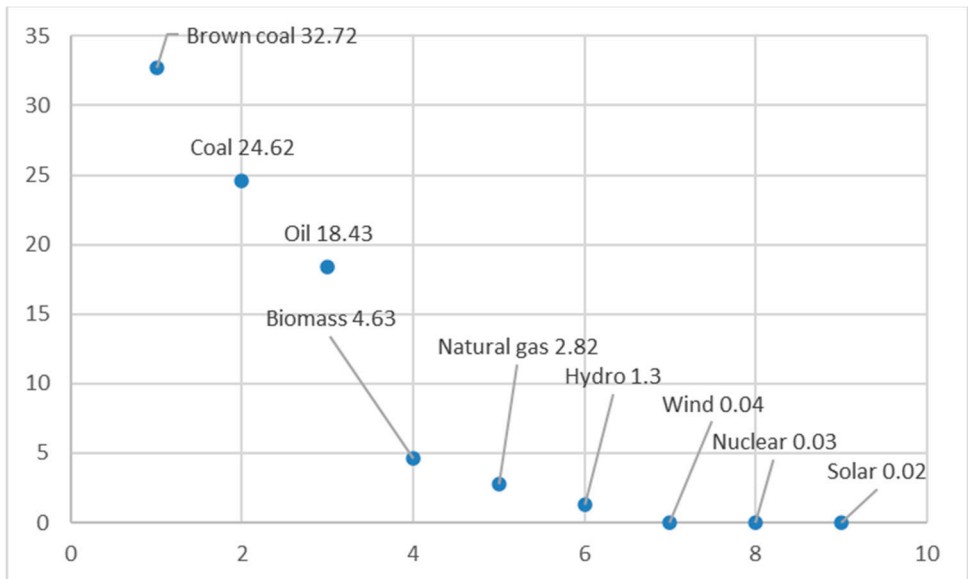

**Figure 11.** Global mortality rates from accidents and air pollution per unit of electricity worldwide by energy source (deaths/thousand TWh), Source of data: [76].

Earlier calculations were offered by Conca, who combined actual deaths and epidemiological estimates to form a *deathprint* indicator that makes it possible to compare different types of energy generation by kWh [77]. In addition to global figures, a comparison for coal was made between US and China, and a comparison for hydro was made between global average and US average to underline some country differences. According to this study, nuclear and hydropower are the safest, followed by global average for nuclear and wind (Table 6).

**Table 6.** Mortality rates associated with different energy sources use.

| Energy Source | Mortality Rate (Deaths/Trillion kWh) |
|---|---|
| Coal—global average | 100,000 (41% global electricity) |
| Coal—China | 170,000 (75% China's electricity) |
| Coal—U.S. | 10,000 (32% U.S. electricity) |
| Oil | 36,000 (33% of energy, 8% of electricity) |
| Natural Gas | 4000 (22% global electricity) |
| Biofuel/Biomass | 24,000 (21% global energy) |
| Solar (rooftop) | 440 (<1% global electricity) |
| Wind | 150 (2% global electricity) |
| Hydro—global average | 1400 (16% global electricity) |
| Hydro—U.S. | 5 (6% U.S. electricity) |
| Nuclear—global average | 90 (11% global electricity with Chernobyl and Fukushima) |
| Nuclear—U.S. | 0.1 (19% U.S. electricity) |

Source of data: [77].

Another estimation with the same conclusion is offered by Hesthamer [78] (Table 7). The study confirms that solar, wind, and nuclear power cause relatively few deaths per unit of generated electricity, while biomass has less favorable characteristics. Fossil fuels are associated with a much higher deathprint and GHG emissions, nuclear being the safest of all traditional fuels, followed by natural gas, which accounts for several times less deaths than biomass.

**Table 7.** Estimated deaths and GHG gas emissions per unit of generated electricity by fuel type.

| | Deaths/TWh | GHG Emissions ($CO_2$eq./kWh) | Share in Primary Global Energy Consumption 2018 (%) |
|---|---|---|---|
| Wind | 0.04–0.15 | 4–11 | 0.8 |
| Nuclear | 0.01–0.07 | 4–12 | 1.7 |
| Biomass | 4.6–24 | 98–230 | 7.1 |
| Solar | 0.02–0.44 | 6–48 | 0.4 |
| Hydropower | 0.02–1.4 | 24–97 | 2.7 |
| Natural gas | 2.8–4 | 490 | 24.5 |
| Oil | 18.4–36 | 715 | 34.5 |
| Coal | 28.7–100 | 820 (lignite 1150) | 27.9 |

Sources of data: [78].

As for other indicators, the cross-country variations are significant. On the one hand, high- and middle-income countries have higher carbon intensity, while carbon emitted per capita has a significantly impact on infant mortality. However, infant mortality is lowest in high-income countries, possibly due to the compensational effect of public healthcare. On the other hand, low-income countries have higher rates of renewable energy adoption (despite advanced technological development in high-income nations) and higher levels of infant mortality. Additionally, it was revealed that as real per capita income goes up, the positive association between renewable energy and infant and child mortality decreases in high- and middle-income countries [79]. Policies that stimulate the reduction in GHG emissions and promote renewable energy reduce deaths associated with air pollution in middle-income countries [80].

## 4. Discussion and Conclusions

This study offers a review and a critical assessment of the existing research and analytical publications that contain comparable data (quantitative indicators) on the contribution of various energy technologies to climate resilience. As a result of this assessment, a framework was suggested to compare low-carbon technologies in line with the UN vision of climate resilience that is supported by the three relevant concepts—energy trilemma, sharing economy/material footprint, and PHDI. The literature review allowed for the identification of suitable indicators for which the data are available: the TMR, the present, and the future levelized cost of electricity (LCOE) without subsidies, $CO_2$ emissions by fuel or industry, lifecycle $CO_2$-equivalent emissions, and mortality rates from accidents and air pollution per unit of electricity worldwide by energy source. These indicators make possible the cross-country and multiple technology comparisons that are particularly relevant for decision-makers in international organizations and companies. The data collected in this study shows that solar, wind, geothermal, tidal, large and small hydropower, nuclear, and other low-carbon technologies vary in their lifetime costs, the amount of greenhouse gas emissions per kWh, air pollution and related health implications, as well as the amounts of required materials and minerals.

This article contributes to the string of research on the interlink between energy industry and climate resilience. Unlike most previous studies that focus on the negative impacts of climate change on energy infrastructure and ways of increasing the resilience of energy infrastructure to climate phenomena [22–24], this paper analyses the positive contributions of low-carbon energy technologies to climate resilience. This study offers a new set of criteria (with relevant indicators) for decision-makers when prioritizing the support of low-carbon technologies and planning the designs of energy systems.

There are not many quantitative studies that suggest measures for various parameters of energy technologies in the context of climate resilience, making it possible to compare these technologies by unit of output or during the entire lifecycle. Those studies that assess the contribution of energy technologies to climate resilience focus on one technology [28] or selected climate resilience aspects, such as material consumption [31] or greenhouse gas

emissions [32]. This study attempts to cover all aspects of climate resilience, as defined by the United Nations, offers a common approach to compare different low-carbon technologies, and reviews existing relevant data. It is somewhat similar to comprehensive studies put forward by international organizations, including the integrated lifecycle assessment of electricity sources by UNECE with a focus on carbon neutrality [35] that is conceptually close to the present review paper. Although similar, the UNECE study is different in terms of its wider thematic focus (environmental impact assessment of various power generation technologies, including resource and material use, land occupation, freshwater eutrophication, human toxicity, and other) and the measurement methods of certain indicators that are expressed in points (for example, lifecycle land use), which limits the objectivity of analysis.

The first and most important conclusion of this study is that renewables, natural gas with CCUS, and nuclear contribute the most to climate resilience. The present cost of electricity (LCOE without subsidy) is lowest for solar PV-utility scale and gas-combined cycle power plants. The best cost projections are for wind onshore and solar that are expected to reach LCOE 10–20 USD per MWh by 2050. Material consumption is lowest for wind and nuclear energy (particularly, closed nuclear fuel cycle) compared with those low-carbon energy technologies for which the comparable data were available.

Average lifecycle $CO_2$-equivalent emissions are lowest for wind, hydro, and nuclear, followed by solar, geothermal, and biomass energy. Without CCS, the lifecycle $CO_2$ emissions of natural gas are approximately 35% higher than large hydro, which is the largest emitting renewable technology. Nuclear power plants have low $CO_2$ emission values comparable to renewables. In addition, excessive power generated at nuclear power plants at times of low demand (for example, at night) may be used for new energy intensive products and services: hydrogen production, supercomputers, data storage, and cryptocurrency mining.

Global mortality rates from accidents and air pollution per unit of electricity are lowest for solar, nuclear, and wind energy followed by hydropower. The lack of harmonized data for this indicator is a significant barrier for comparison with other technologies. The deathprint of biomass is significantly higher compared to other renewables, while this indicator for natural gas is lowest of all fossil fuels.

Natural gas with CCUS and nuclear along with storage systems could complement well variable renewables, thus increasing the reliability of low-carbon energy systems. Natural gas-combined cycle with CCUS at present is cost competitive with utility-scale installations combining onshore wind or solar PV with storage. It will not change significantly after 2030 and by 2050 will amount to USD 53–118 in the Sustainable Development scenario. The LCOE of advanced nuclear technologies will level off at USD 60–110 after the year 2030 in the Net-zero scenario, making it much more cost competitive with natural gas than it is at present. Despite the research and policy debates about long-term sustainability of natural gas and nuclear, nearly all forecasts project various shares of these fuels in the 2050 global energy mix. Given the urgent demand for decarbonization in line with the Paris Agreement and the high input of the energy sector in global greenhouse gas emissions, it seems relevant to rely not only on renewables (except for first and second generation biomass), but also on natural gas with CCUS and advanced nuclear technologies.

High-income countries are in a better position in terms of disposable income, technological base, and the ability to assure the safe use of energy resources or compensate for existing negative effects. Low-income countries, despite the relatively higher pace of low-carbon technology adoption, face more severe negative consequences of climate change and are less climate resilient. This may be illustrated with high mortality rates from air pollution and low availability and affordability or renewable energy in low-income countries.

This review revealed that little data are gathered on non-mainstream renewables and new energy sources including tidal and wave energy, hydrogen, and even geothermal energy. The TMR for renewables is much better studied than for hydrocarbons and nuclear. The use of data from different studies and sources is often not possible due to different

research methods and approaches that are not harmonized. It should also be noted that this review focuses on global average data, while the estimations may vary significantly from country to country and even at the subnational level. These assessments are also technology specific and, given the high pace of technology advancement, need to be revisited on a regular basis.

**Supplementary Materials:** The following supporting information can be downloaded at: https://www.mdpi.com/article/10.3390/cli11120231/s1, Table S1: Keywords used for data and information search.

**Funding:** The article was prepared in the framework of a research grant provided by the Ministry of Science and Higher Education of the Russian Federation (grant ID: 075-15-2022-325).

**Data Availability Statement:** Data sharing not applicable to this article as no datasets were generated or analyzed during the current review study.

**Conflicts of Interest:** The author declares no conflict of interest. The funders had no role in the design of the study; in the collection, analyses, or interpretation of data; in the writing of the manuscript; or in the decision to publish the results.

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
