# Peer review of "The Contribution of Low-Carbon Energy Technologies to Climate Resilience"

_climate, doi:10.3390/cli11120231_

Round 1
Reviewer 1 Report
Comments and Suggestions for Authors
This review paper introduces the present progress in energy technologies to climate resilience. Overall it’s well-organized and clearly-stated. But there are still some improvements can be made.
I suggest the author to polish the abstract with a focus on background, objectives, methodology, main findings and conclusion. Please add a sentence which shows the necessity of the study. In the introduction part, please follow the literature review by a clear and concise state of the art analysis. You should reason both the novelty and relevance of your paper goals. Please revise the conclusions. Conclusions are not just summarizing the key results of the study, it should highlight the insights and the applicability of your findings for further work.And for the reference, I think you should put more articles from top journals as references.
Comments on the Quality of English LanguageLine 43 is lack of a full stop. Line 161 has a redundant “shown”. You should pay more attention to the language details, especially for those grammatical errors and bad structured sentences. The language can be further improved by native speakers.
Author Response
Dear Reviewer,
Thank you for your time and suggestions on how to improve the paper. They are much appreciated. Attached are the detailed responses to your comments.
Best regards,
The author

Reviewer 2 Report
Comments and Suggestions for Authors
Review Report
“The contribution of low-carbon energy technologies to climate resilience”
Summary: The article reviews the means for comparing different energy sources on basis of their resilience, accessibility, affordability and contribution to pollution. The concepts introduced in this paper would be useful for people of varying backgrounds including academics, policy makers and private sector companies.
It is recommended that the article be considered for publication in this journal with major revisions. The content of the manuscript presented is sound. The content of this manuscript is within the scope of the journal. The manuscript has a sturdy framework from research conducted. However, major changes are required in terms of the papers considered for this manuscript, as it is a broad topic, multiple different sources of information would exist. Further exploring the combinatory effect of these factors should be explored. In addition alternating measurements techniques for comparing energy technologies should be atleast introduced.
General Comments:
· The abstract should include a brief compilation (one/few sentence for abstract) of the most important findings from the review.
· Section 2 needs to further explain the correlation between “The UN approach to climate resilience” and the components explored in this paper. Why are they related and how is one similar to the other/ act as a proxy.
· Sufficient papers have not been referenced in the study. More studies related to each topic are required for this manuscript (example: only 5 papers have mentioned in section 3.1). Also mentioning statements shown in line 183-185, “Of all studies published” without giving an indication of which papers are being referenced, isn’t suitable for a review article.
· A combinatory study using all three indicators should be explored or atleast introduced. Especially the factor related to weighing the importance of these; maybe a basic analysis using equal importance provided to each indicator could give a preliminary understanding.
· The graphs in the manuscript need to be cleaned. Example: Fig. 7.
· Literature developed on alternate means of comparing the energy sources should also be explored/introduced in the study.
Comments on the Quality of English LanguageThe quality is good. The manuscript grammar requires some minor work.
Author Response

(The authors gave the same response as above.)

Reviewer 3 Report
Comments and Suggestions for Authors
Dear Authors,
You did a great work and the paper “The contribution of low-carbon energy technologies to climate resilience” is really interesting, but in the current form it need some moments to revise in order to be published in Climate.
I would like to highlight main issues that must be taken into account.
Also, there are following comments for your attention:
Introduction section should prove the relevance and the importance of your study. Introduction need to be strengthened with figures and data in order to provide the actuality of the goal. From this section it should be clear what is the goal of the research and why it is important.
You wrote “This paper aims to review and analyze the existing pool of studies published 63 by researchers and international organizations that offer comparable data (quantitative 64 indicators) on the contribution of various energy technologies to climate resilience”
So if your paper is review it should be methods that you used to analyze existing study. How did you do it?
It is not clearly understandable how methods interconnected with the achieved results. It should be clearly highlighted what is contribution of low carbon technologies to three outcomes of climate resilience.
Do you used described PHDI concept in your results?
Literature review for review article is extremely important in my opinion but it is absent in the paper. Also maybe some systematization of approaches and methods is needed.
I suppose some conclusions after table 1 are needed.
Sources should be mentioned for all tables in the paper.
Conclusions section should be strengthened in order clearly present results and achieved goals of the research.
The idea of the paper is really good and I suppose it could be presented in more appropriate way.
I hope my comments will be useful and will help you to improve the paper.

Author Response

(The authors gave the same response as above.)

Reviewer 4 Report
Comments and Suggestions for Authors
Your paper entitled "The contribution of Low-carbon energy technologies to climate resilience" is set with a high ambition, and it gathers substantive information that can be of great interest for readers interested in discerning what energy technology contributes the most to climate resilience and how do different technologies compare. However, there are concepts and considerations that need to be explained clearly before the paper can be accepted for publication.
- The title: it refers to "low-carbon technologies", you will need to clarify from early in the paper that you are including here, not only renewable energy technologies, but also high carbon technologies with use of carbon sequestration, storage or utilization CCS or CCUS. Adding CCS or CCUS may make the cost comparison on the contribution of these technologies to climate resilience vary greatly and the question of context cannot escape this analysis. Please explain how do you apply the concept of "low carbon technologies" in your analysis. Particularly, clarify pag2 lines 53-56
-The concept of climate resilience is unclear. You depart from the UN approach, that situates resiliency in people, businesses and the environment as giving the meaning of climate resilience. Your goal is to create a framework of indicators that can help measure the technologies contribution toward climate resilience. However, it all reads very ambitious and vague. From environmental perspective, does climate resilience means the Paris Goals and reducing emissions to keep temperatures under 1.5 oC?s There is a risk of trivializing or diluting climate mitigation. You do not make clear what the goal is with climate resilience. Regarding people what resiliency of people really means? If it is energy affordability to the poorest people, then you need to keep this in mind when comparing what energy technologies contribute the most to climate resiliency.
Pag2-Lines 73-38: Here is the opportunity for you to clarify the extent to which environmental resiliency means the same as meeting global climate goals. I doubt that something can have the title of "climate resilience" when it only means addressing water and air pollution. what about carbon emissions?
Pag 2 Lines 85-87: the energy trilemma paragraph needs to be re-written. Specially the last lines. It either contributes to enhance growth and environmental quality in the long run or is negatively correlated with environmental quality. Unclear. Please revise
Pag 3 lines 92-102 claims are made about the contribution of sharing economy, circular economy to a potential future of less material consumption and use. These cannot be presently attributed to any of the "low carbon technologies" you are seeking to compare. Aside from disclosing the assumed potential that sharing and circularity will have, it is unclear how do you see these impacting or being part of the implementation of "low carbon technologies". If you are seeking here to connect circularity with carbon storage and utilization, this will be a good place to make such connections. There is a lot that could be said in terms of energy efficiency and energy demand management to reduce consumption, which one could associate with some of the R's in the circular economy. Please expand or try to be more specific.
Page 5-lines 122-127 Finally you come to the core of the indicators that you will be applying as a "framework" they are: "present and future energy affordability, GHG emissions and environmental pollution and material footprint." How these indicators support elements of the energy trilemma, or elements of sharing economy and circularity and PHDI? Are the chosen indicators not affected by context, geography, economic development? Despite the summary made in Table 1, more explanations are required.
Page 7 lines 187-188: finally two lines indicating energy efficiency can play a role linked to sharing economy and reduction of material used. I will encourage you to expand these two lines with the more challenging proposition that resiliency demands a level of demand (in terms of material energy and emissions) reduction that will make possible to handle the use of resources needed for the energy transition.
Page 10 lines 242-246 a stronger conclusion is needed about LCOE's results. Particularly, because you are considering a strong role for carbon sequestration and utilization.
Page 12 lines 290-291 in passing is introduced CCSU, average life cycle CO2 equivalent emissions per KWh for natural gas and renewables becomes comparable. Yes, however there is a cost that is not discussed here. In addition a short/long term comparison that is not explained. Renewable energy technologies can deliver the cheapest and cleanest electron today, Cannot be left unexplained that CCSU is not a technology that is presently helping natural gas emissions be comparable to those of renewables on a lifecycle perspective.
Page 15 lines 357-358- you claim that the most important conclusion is that renewables, natural gas with CCUS and nuclear have the best characteristics for climate resilience. Given your own results a stronger conclusion will favor renewables and nuclear with caveats and reservations toward natural gas as it depends on CCUS which is not currently available.
Overall, there needs to be a clearer alignment of what "climate resilience" means in terms of carbon budget and contribution to keeping warming under 1.5 degrees C. There cannot be "climate resilience" achieved in a warmer than 2 degree planet. The paper does not deal sufficiently well with this critical condition of climate resilience.
Comments on the Quality of English Language
The paper will benefit from having an English editing review
Author Response

(The authors gave the same response as above.)

Round 2
Reviewer 3 Report
Comments and Suggestions for Authors
Dear Authors,
You did a great work, but the paper still need some clarifications, cause there is some kind of a mess.
The paper is claimed as a Review paper. At the same time there are a lot if data and tables with quantities. What is the table 3? How it contribute the goal of the paper?
The Authors claims that their “paper aims to review and critically assess the existing pool of studies published by researchers and international organizations that offer comparable data (quantitative indicators) on the contribution of various energy technologies to climate resilience”.
Did the authors achieve the goal? The authors in the conclusion said that the paper suggests “a framework to compare low-carbon technologies in line with the UN vision of climate resilience that is supported by the three relevant concepts – energy trilemma, sharing economy/material footprint, and PHDI”.
Please organize the research design in order to track connection between goal-methods-results- conclusions.
Author Response
Dear Reviewer, thank you for taking your time to review the article for the second time. Attached are more detailed responses to your comments.

Reviewer 4 Report
Comments and Suggestions for Authors
Thank you for reviewing the paper and responding positively to made additions and to deal with some of the suggestions and issues raised. I have now positively recommended your paper be accepted for publication.
Author Response
Dear Reviewer,
Thank you for taking your time to review the article for the second time and thank you for your positive assessment of the implemented revisions.
Best regards,
The author

Round 3
Reviewer 3 Report
Comments and Suggestions for Authors
I would suggest to authors to add some disscussion in conclusion section
Author Response
Dear Reviewer, thank you for your suggestion. Following your and Editor's comments, the Conclusions section was renamed into Discussion and Conclusions and now includes similarities and differences of the review paper with other thematically close studies (2nd and 3rd paragraphs were added).